# The Potential Effect of General Anesthetics in Cancer Surgery: Meta-Analysis of Postoperative Metastasis and Inflammatory Cytokines

**DOI:** 10.3390/cancers15102759

**Published:** 2023-05-15

**Authors:** Ru Li, Mousumi Beto Mukherjee, Zhaosheng Jin, Hengrui Liu, Kevin Lin, Qiuyue Liu, James P. Dilger, Jun Lin

**Affiliations:** Department of Anesthesiology, Stony Brook University Renaissance School of Medicine, Stony Brook, NY 11794-8480, USA

**Keywords:** sevoflurane, isoflurane, desflurane, inhalational anesthesia, propofol, total intravenous anesthesia, cancer surgery, metastasis, recurrence, IL-6, TNF-α, IL-10

## Abstract

**Simple Summary:**

This meta-analysis examined the effect of general anesthetics on metastasis and recurrence after cancer surgery from clinical and pre-clinical studies. It showed that propofol-based total intravenous anesthesia is associated with lower risk of metastasis/recurrence and lower IL-6 level than inhalational anesthesia. Pre-clinical studies confirmed clinical observation and explored potential mechanisms.

**Abstract:**

Metastasis or recurrence following curative surgery is the main indicator of tumor progress and is the main cause of patient death. For more than three decades, the potential for general anesthesia to affect cancer outcomes has been a subject of concern with considerable research interest. Here, we conducted this systematic review and meta-analysis to summarize the effect of inhalational anesthesia (IHNA) vs. propofol-based total intravenous anesthesia (TIVA) on metastasis and recurrence after cancer surgery from clinical and pre-clinical studies. The relative risk for metastasis/recurrence in TIVA is 0.61 (95% confidence interval (95% CI) 0.46 to 0.82, *p* = 0.0009) compared to IHNA. Inflammatory cytokines have been implicated in cancer metastasis following cancer surgery, thus we analyzed inflammatory cytokines levels after surgery under IHNA or TIVA. Based on pooled analysis, a lower IL-6 level was noticed in TIVA in comparison to IHNA (standardized mean difference (SMD) = 0.77, 95% CI = 0.097 to 1.44, I^2^ = 92%, *p* = 0.02) but not TNF-α or IL-10. Preclinical animal model studies show that inhalational anesthetics increase the risk of breast cancer metastasis compared to propofol. In conclusion, the current evidence suggests intravenous anesthetic propofol is associated with less metastasis/recurrence and lower postoperative IL-6 level over inhaled anesthetics in the oncological surgery. We urge more well-designed clinical and preclinical studies in this field.

## 1. Introduction

Surgery is one of the first-line treatments for millions of patients with solid tumors each year. However, metastatic recurrence in vital organs remains as a main cause of death. Significant advances have been achieved in surgical techniques such as minimally invasive and robotic-assisted approaches to reduce surgical trauma. Recognition of the modifiable factors during the critical perioperative period is essential to reduce lethality of recurrent or metastatic diseases. It has been shown that surgery and the accompanying anesthesia induce profound inflammatory, immunological, and metabolic stress on cancer patients that may accelerate local tumor recurrence and distant metastasis [1]. Thus, the choice of better general anesthetics may prove to be an effective approach to improve the long-term outcome of cancer patients and is increasingly seen as a research priority.

In current practice, general anesthesia is usually induced with an intravenous anesthetic, mostly propofol. General anesthesia is maintained with either continuous infusion of propofol (total intravenous anesthesia, TIVA) or an inhaled anesthetic such as sevoflurane, isoflurane, or desflurane (inhalational anesthesia, IHNA). The choice of general anesthetics is usually the decision of the anesthesiologist based on the co-morbidity and condition of the patient and the preference of the anesthesiologist. General anesthetics have been shown to have significant effects on the biology of cancer cells, and their role in clinical outcomes has been a concern for several decades, underpinning the emerging field of onco-anesthesia [2,3,4,5]. For example, sevoflurane has been shown to increase cancer cell resistance to chemotherapeutic agents and increase cell migration and viability of renal carcinoma cells but reduce viability of non-small cell lung carcinoma cells. Propofol has been shown to decrease cell proliferation and migration in various cancer cells including pancreatic cancer, ovarian cancer, hepatocellular carcinoma, lung adenocarcinoma, and breast cancer. Numerous, mainly retrospective, trials have been conducted over the years to analyze the effect of general anesthetics on the long-term survival of cancer patients, and several meta-analyses have been published. Recurrence or metastasis is an objective indicator of disease progress. Metastasis to the distant organs is the main cause of patient death and often an endpoint in animal studies. In our previous meta-analysis, we found that TIVA is associated with longer overall survival (OS) and recurrence-free survival than IHNA [6]. To dig deeper into the data, we conducted a systematic review and meta-analysis to investigate the effect of commonly used general anesthetics on cancer metastasis. In addition, general anesthetics have been shown to alter inflammatory cytokines secretion in breast cancer, which contributes to post-surgery inflammatory responses [7]. The change in postoperative cytokine profiles may also increase the risk of infectious complications and affect lung cancer metastasis after surgery [8]. Therefore, the second objective of this paper is to perform the meta-analysis of the effect of general anesthetics on cytokine production after cancer surgery. To identify plausible mechanisms that underlie the impact of general anesthetics on cancer metastasis, researchers are looking concurrently to pre-clinical animal studies for answers. These studies may identify potential targets for clinical intervention. Thus, the third objective of this paper is to perform an overview of animal studies. Our goal is to summarize all the available information about the ability of general anesthetics to affect metastasis.

## 2. Materials and Methods

### 2.1. Search Strategy

This study conformed to the Preferred Reporting Items for Systematic reviews and Meta-Analysis (PRISMA) statement [9] and was registered on PROSPERO (CRD42023410052). For the analysis of cancer metastasis, as shown in Table 1, we used search terms ‘TIVA’, ‘total intravenous anesthesia’, ‘propofol’, ‘inhaled anesthesia’, ‘volatile anesthesia’, ‘cancer’, ‘tumor’, ‘malignancy’, ‘neoplasm’, ‘recurrence’, ‘metastasis’, and their Boolean combinations. For the analysis of postoperative inflammation, we used search terms ‘TIVA’, ‘total intravenous anesthesia’, ‘propofol’, ‘inhaled anesthesia’, ‘volatile anesthesia’, ‘cancer’, ‘tumor’, ‘malignancy’, ‘neoplasm’, ‘cytokine’, ‘interleukin’, and their Boolean combinations. All the searches were conducted in PubMed, Central, EMBASE, CINAHL, Web of Science citation index, US clinical trials register, Google Scholar, UK clinical trials register, Australia, and New Zealand clinical trials register. We did not restrict any language at the time of search. All literature searches were conducted by two authors independently, and discrepancies were discussed afterwards.

### 2.2. Study Eligibility Criteria

For cancer outcome analysis:All randomized-controlled trials (RCT) and observational longitudinal studies (prospective and retrospective) comparing metastasis and recurrence after surgery with IHNA or TIVA were included.Studies reporting metastasis incidence, recurrence incidence, or recurrence rate were included.

For cytokine analysis:All randomized control trials in adult patients undergoing surgery under general anesthesia of IHNA or TIVA.Studies reporting at least one of the cytokines, IL-6, IL-10, or TNF-α.Studies including comparisons expressed as mean ± standard deviation or comparison values represented as median.

Exclusion criteria were studies that did not include data in a suitable format, studies with children under 18, and ongoing clinical trials.

### 2.3. Data Extraction

Data extraction was conducted based on standardized proforma and double-checked by a second author (RL, MBM, and ZJ). Extracted data included bibliographical information (author and year), study design (prospective or retrospective study, number of patients in the IHNA and TIVA cohort, cancer type), and the outcomes (metastasis or recurrence, inflammatory cytokines).

We employed the Quality of Prognostic Studies (QUIPs) tool to evaluate the quality of the included studies. The QUIPs tool is a 6-item questionnaire designed for assessing both prospective and retrospective studies. Each item as a risk category can be determined to be low, medium, or high risk [10]. All assessments were performed by two authors independently at the same time, and any disagreement was discussed with and resolved by a third author (JL).

For those studies, when results were displayed only as graphical form, data were extracted using WebPlot Digitizer [11]. For the studies where the plasma cytokine values were expressed as ‘medium’ and no standard deviation values were reported, the mean ± standard deviation values were obtained using ‘Deep Meta tool Version 1.0’ [12]. This tool utilized Bland’s method when the entire data set was reported, but Hozo’s method was used in cases when only the median and interquartile range were stated.

### 2.4. Statistical Analysis

Meta-analysis was conducted for outcomes reported in more than two studies. We used Review Manager (RevMan) Version 5.4 (Copenhagen: The Nordic Cochrane Centre, The Cochrane Collaboration, 2014) for the pooled analysis. For metastasis or recurrence, we calculated relative risks with 95% CI for dichotomous outcomes by the Mantel–Haenszel method (fixed or random models). For overall survival, the pooled hazard ratio (HR) of TIVA against INHA was calculated from the HR of each studies using generic inverse variance method with a 95% confidence interval (CI) [13]. In studies that did not report the hazard ratio, HR was estimated using methods described by Tiemey et al. [14]. For post-operative inflammatory cytokines, continuous outcomes were expressed as a mean value and standard deviation and were analyzed by using standard mean difference. Heterogeneity was assessed using Cochrane’s I^2^ statistic, expressed as a percentage term; higher percentage suggests higher degree of heterogeneity [15]. According to the Cochrane review guidelines, if severe heterogeneity was present at I^2^ > 50%, the random effect models were chosen, otherwise the fixed effect models were used. Potential publication bias was detected by using the funnel plot and Egger’s regression (statistical significance indicates high probability of publication bias, *p* < 0.05 is considered significant likelihood of bias) with statistical package provided by Suurmond et al. [16,17]. Subgroup analyses were conducted for those data which were collected from metastasis against recurrence studies and for organ involved.

## 3. Results

### 3.1. General Anesthetics and Cancer Metastasis

Here, we summarize the currently available findings from clinical studies on metastasis or recurrence after cancer surgery under different general anesthetics. Recurrence can be local or distant (metastasis). Here, we consider local recurrence as a form of metastasis for the following reasons. In clinical practice, the primary tumor is typically completely resected with clean margins. Radiation therapy is usually used to destroy residual cancer cells in the local environment. Therefore, local recurrence most likely arises from the seeded circulating cancer cells. Thus, considering both recurrence and metastasis together may increase the power of our meta-analyses.

Our systematic search on PubMed, Central, EMBASE, and CINAHL identified 20 studies to be included in this meta-analysis. The PRISMA flow diagram of the study selection is shown in Appendix A, and the study characteristics are summarized in Appendix A. There were three prospective studies and 17 retrospective studies. There were eight studies on breast cancer, three on liver cancer, and two on brain cancer. There was one study each for gastric, pancreatic, prostate, oral, bladder, bone, and gynecologic cancer. The risk of bias assessment for each study is presented in Appendix A.

Of the 20 included studies, the rate of metastasis or recurrence (or recurrence incidence) was reported for patients matched with a propensity score system (Figure 1) [18,19,20,21,22,23,24,25,26,27,28,29,30,31,32,33,34,35,36,37]. Patients in the TIVA cohort had a highly significant lower incidence of metastasis or recurrence than those in the INHA cohort with a relative risk (RR) of 0.73 (95% confidence interval (95% CI) 0.62 to 0.86, I^2^ = 76%, *p* = 0.0001). The funnel plots for publication bias are presented in Appendix A. To eliminate the confounding factors related to retrospective studies, we conducted a subgroup analysis between perspective and retrospective studies (Figure 2). For the three RCTs, patients in the TIVA cohort had lower risk of metastasis/recurrence than patients in the IHNA cohort, but the difference was not significant (RR = 0.63, 95% CI 0.27 to 1.47, I^2^ = 39%, *p* = 0.28). For retrospective studies, significantly lower risk of metastasis/recurrence was observed in patients from the TIVA cohorts. To distinguish between metastasis and recurrence, we conducted a subgroup analysis (Appendix A). For metastasis, patients in the TIVA cohort were associated with a highly significant lower risk than patients in IHNA cohort (RR = 0.67, 95% CI 0.58 to 0.77, I^2^ = 23%, *p* < 0.00001). For recurrence, there was no significant difference between the TIVA and IHNA (RR = 0.90, 95% CI 0.82 to 1.00, I^2^ = 65% *p* = 0.05). Due to the moderate heterogeneity (I^2^ = 76%) amongst the studies, we also conducted a subgroup analysis by types of cancer (Appendix A). For breast cancer, the TIVA group had a small but significant lower incidence of metastasis or recurrence than the IHNA group (RR = 0.87, 95% CI 0.77 to 0.99, I^2^ = 26%, *p* = 0.04). For liver cancer, the TIVA group was also associated with lower risk of metastasis or recurrence (RR = 0.65, 95% CI 0.49 to 0.87, I^2^ = 0%, *p* = 0.004). There was no significant difference between TIVA and INHA in patients with brain cancer, which were not yet conclusive due to limited sample size (Appendix A).

Of the 20 included studies, 16 reported risk estimates for overall survival (OS) between TIVA and INHA [18,19,20,21,22,23,24,25,26,27,28,29,30,31,32,35]. Based on meta-regression, patients receiving TIVA were associated with better overall survival than the patients who received IHNA (hazard ratio (HR) = 0.75, 95% CI 0.63 to 0.90, I^2^ = 76%, *p* = 0.002) (Figure 3). This mirrored the pooled RR of metastasis/recurrence in Figure 1.

### 3.2. General Anesthetics and Inflammatory Cytokines

It is widely accepted that both cancer recurrence and metastasis are significantly influenced by immunologic function changes during surgery. Cytokines are considered to be mainly secreted from the surgical wound and concurrently released into the bloodstream. General anesthetics have been suggested to affect cytokine production, which, in turn, affects the function of immune cells. Therefore, we analyzed the post-operative levels of inflammatory cytokines in cancer patients treated with different anesthetics. Through searching on PubMed, Central, EMBASE, and CINAHL, we included nine studies in the meta-analysis; the PRISMA flow diagram of the study selection is shown in Appendix A. Study characteristics were summarized in Appendix A, and the risk of bias assessments are in Appendix A.

There were nine studies that reported plasma levels of IL-6 24 h after surgery [38,39,40,41,42,43,44,45,46]. These included a total of 285 patients who received IHNA and 282 patients who received TIVA. The pooled results showed statistically lower IL-6 level in the TIVA cohort in comparison to the INHA cohort (Figure 4, standardized mean difference (SMD) = 0.77, 95% CI = 0.097 to 1.44, I^2^ = 92%, *p* = 0.02). The quality of evidence was low due to significant data heterogeneity (I^2^ = 92%). A subgroup analysis based on cancer surgery type was also conducted in order to address the heterogeneity of the study where 255 and 252 patients were in the INHA and TIVA groups, respectively. This included eight studies of three different types of cancer (breast cancer, lung cancer, and esophageal cancer). The analysis revealed that the IL-6 levels in each cancer type were not significantly different between different anesthesia groups (Appendix A). There were four studies that reported serum TNF-α level 24 h after surgery [39,40,42,45]. This study involved a total of 88 patients who received IHNA and 88 patients who received TIVA. The pooled results represented no significant change in both anesthesia groups (SMD = −0.20, 95% CI = −0.47 to −0.07, I^2^ = 95%, *p* = 0.15, Appendix A).

Six studies reported on serum IL-10 level 24 h after surgery [38,39,40,43,45,46], in which 218 patients received INHA anesthesia and 221 TIVA anesthesia. The pooled results from these studies showed no significant differences in IL-10 levels (SMD = −4.93, 95% CI = −30.27 to 20.40, I^2^ = 95%, *p* = 0.70, Appendix A). Considering the small number of studies and the heterogeneity of surgical procedures [47], it is not clear if TIVA has a significant effect on the expression of regulatory cytokines.

### 3.3. General Anesthetics and Cancer Metastasis in Pre-Clinical Animal Studies

Pre-clinical models are powerful tools to identify potential risks and refine the treatment approach. Pre-clinical models are also useful in identifying molecular mechanisms that underlie the impact of anesthetic agents on cancer metastasis. Here, we summarize pre-clinical animal studies of general anesthetics on cancer metastasis (Table 2). Three animal studies compared different general anesthetics in the context of surgery. Our group found that surgical resection of primary tumors under sevoflurane anesthesia led to significantly more lung metastasis than that with propofol anesthesia [48]. This distinction was observed in two mouse models: the murine 4T1 syngeneic mouse breast cancer model and the human MDA-MB-231 breast cancer xenograft mouse model. The underlying mechanism mediating the effect of sevoflurane was via the increased production of IL-6 that activated its downstream transcription-factor STAT3 in mouse lungs. Moreover, sevoflurane increased the infiltration of CD11b+ myeloid cells in the lungs; these cells orchestrated the microenvironment for the growth of metastatic tumors. We further showed that the commonly used inhaled anesthetics, namely, isoflurane, sevoflurane, and desflurane, did not show any difference between each other on tumor cell growth in vitro or lung metastasis following surgery in the 4T1 model [49]. Freeman et al. used propofol and lidocaine in combination with sevoflurane on postoperative breast cancer lung metastasis in the 4T1 syngenetic mouse model incorporating surgery. The addition of propofol to sevoflurane reduced postoperative pulmonary metastasis compared to sevoflurane alone but did not affect hepatic metastasis nor serum IL-6 and VEGF at five weeks after surgery.

Four animal studies examined a single general anesthetic agent without a surgical procedure, which might limit their clinical relevance. Two of them investigated the effects of propofol on cancer metastasis. Mammoto et al. employed the subcutaneous inoculation of murine osteosarcoma (LM 8) cells in mice to test the effect of continuous infusion of propofol or vehicle for four weeks [50]. The infusion of propofol did not affect the primary tumor growth but significantly decreased pulmonary metastasis. It should be noted that the dose of propofol (continuous infusion of 20 or 40 mg/kg per day) did not induce significant sedation nor anesthesia. Liu et al. used an experimental metastasis model of tail vein injection of human colorectal HCT116 cells under propofol anesthesia versus non-anesthesia conditions [51]. They found that propofol significantly increased the formation of metastatic lung tumors. The mechanism proposed in this study was that propofol enhanced circulating tumor cell adhesion by GABA_A_R-dependent Src ubiquitination, which was, however, only validated in vitro. Only two animal studies examined the effects of isoflurane on cancer metastasis by utilizing an experimental metastasis model of tail vein injection [52,53]. Isoflurane was shown to increase the pulmonary metastasis of murine B16 melanoma cells (Moudgil 1997) and hepatic metastasis of Human T24 bladder cancer (Lu 2020). The promoting effect of isoflurane on bladder cancer metastasis was suggested to occur through HIF-1α-β-catenin/Notch1 pathways [53].

## 4. Discussion

A body of evidence is emerging that addresses the impact of general anesthetics on cancer progression. Here, we have analyzed the clinical and pre-clinical evidence regarding the influence of general anesthetics on tumor metastasis. Our meta-analysis indicates that intravenous anesthetic propofol is associated with less metastasis and recurrence than inhaled anesthetics. The beneficial effect of the intravenous anesthetic propofol on reducing cancer metastasis is correlated with longer survival and somewhat lower post-operative inflammatory cytokine levels. As the studies included multiple cancer types, we also performed subgroup analysis. Reduced metastasis in TIVA group is only observed in breast cancer but not in other types of cancer. However, the subgroup analysis was limited by the number of studies (2 to 9 studies); more studies are necessary for each cancer type to allow a powered meta-analysis. For the 20 studies included in studying metastasis, there were only three RCTs. The remaining retrospective studies are confounded by the biases inherent to retrospective studies. The limitations of retrospective studies and inconsistencies in reporting preclude the small cohort studies from contributing meaningfully to a high-quality meta-analysis. Therefore, the possibility that general anesthetics differentially affect metastases is very suggestive but must be further investigated.

To answer the question definitively, RCTs as the gold standard of evidence for changing clinical practice are needed to interrogate the clinical impact of general anesthetics on cancer metastasis. Therefore, we conducted the subgroup analysis according to study design. However, no significant differences were observed in the three RCTs. Among the three RCTs included, Sessler 2019 was the major contributor to the final RR. This study recruited 2108 breast cancer patients from multiple hospitals and showed that the combination of intravenous propofol with paravertebral blocks using the local anesthetic ropivacaine did not lower the risk of breast cancer recurrence within a median 3-year follow-up compared to the combination of sevoflurane anesthesia with opioids [34]. The majority of the recruited patients in this trial had estrogen receptor breast cancer with better long-term prognoses. This was also reflected in the low (10%) recurrence rates in both groups. Taking into account the relatively short follow-up time (median follow-up of 3 years), the effect of the general anesthetics on the cancer metastasis/recurrence may have been masked in this prospective clinical trial with large population. The other two RCTs did show significantly lower metastasis and recurrence rates in the TIVA group than the IHNA group. Therefore, we do not think the results of RCT negate the conclusion that TIVA is superior to IHNA in cancer surgery. There are several ongoing RCTs in this area that may shed some light on the topic [6]. It is imperative for future RCTs to consider the multitude of complex clinical factors that may all play a part in the perioperative modulation of cancer surgery.

The surgical manipulation of a tumor is known to induce systemic stress responses, which initiate an acute inflammatory response related to local tissue damage and the shedding of malignant cells into the blood and lymphatic circulation. Our meta-analysis showed that post-operative levels of IL-6 were significantly lower in the propofol group, which indicated the anti-inflammatory effects of propofol. As pleiotropic cytokines, IL-6 and TNF-α initiated the early inflammatory responses to tissue injury, were believed to be important mediators in the systemic response to surgery, and had a high predictive value for the development of post-operative complications. Accumulating evidence illustrated a clinical correlation between post-operative inflammatory cytokines such as IL-6 and metastasis. In a study by Maeda et al., the postoperative serum levels of inflammatory cytokines and tumor metastasis were recorded in patients who underwent radical esophagectomy for primary esophageal cancer. They found that serum IL-6 significantly correlated with the lymph node metastasis and liver metastasis; the serum IL-8 levels also tended to associate with the lymph node recurrence [54]. A similar observation was evident by Shimazaki et al. in colorectal cancer surgery and by Barea et al. in lung cancer surgery, who showed that the postoperative serum levels of IL-6 and TNF-α correlated with distant metastasis [55,56]. Another cytokine we analyzed in this study was IL-10. IL-10 is a regulatory cytokine that downregulates macrophages and nuclear factor kappa-light-chain-enhancers of activated B cell (NF-κB) activities; it also reduces the production of TNF-α, Interferon **γ**, and other proinflammatory cytokines [47]. IL-10 is also the main cytokine involved in promoting wound healing [57]. Overall, targeted interventions to optimize outcomes among vulnerable patients by minimizing perioperative inflammation are a field rich for further studies.

Pre-clinical animal studies have shown that exposure to inhaled anesthetics, such as isoflurane and sevoflurane, can promote the growth and spread of cancer cells, including breast cancer and melanoma. On the other hand, the results of propofol are not conclusive. Several studies observed inhibition effect of propofol on cancer metastasis or less metastasis compared to inhaled anesthetics, while one study showed the promotion effect of propofol on colorectal cancer lung metastasis. These studies agreed with our meta-analysis that more metastases are observed in patients undertaking oncology surgery with inhaled anesthetics. Another benefit of pre-clinical studies is that they could provide us with information especially on molecular mechanisms that we cannot easily retrieve from human studies and identify potential targets for clinical intervention. However, until now, only three studies explored the molecular mechanisms associated with general anesthetics in cancer. Our group found that inhaled anesthetics appeared to increase the number and risk of metastases with or without surgery, while the intravenous anesthetic propofol showed beneficial effects. This echoed the meta-analysis of clinical studies, as shown in Figure 1. In breast cancer lung metastasis, sevoflurane exposure during surgery primed the lung microenvironment via the IL6/JAK/STAT3 pathway. In bladder cancer hepatic metastasis, isoflurane activated the HIF-1α-β-catenin/Notch1 pathways. Different from all the other animal studies, Liu showed that propofol could enhance tumor cell adhesion and extension through GABA_A_R to downregulate TRIM21 expression, leading to the upregulation of Src, a protein associated with cell adhesion. This study compared propofol to control (DMSO) but not inhaled anesthetics in an experimental metastasis model, which made it less relevant to the practical clinic. Although this study suggested potential targets for propofol, most of those results were obtained under in vitro conditions that were not yet clear concerning how or whether this knowledge could be translated clinically. This preclinical animal experimental approach filled the gap between clinical trials and cancer cell biology. It offered well-defined endpoints and a mechanistic study under controlled experimental conditions that mimicked the clinical situation. It is imperative that future in vivo studies are developed using clinically relevant models of spontaneously metastasizing cancer from orthotopic sites. How the results of animal studies translate into effective clinical application is always inconsistent and debated. Nevertheless, it may be prudent to choose anesthesia regimens that have potentially beneficial effects, as advocated by Mao and colleagues in 2013 [58].

An important limitation of this review was its high heterogeneity, as quantified by I^2^ static in each figure. As most of the included studies for metastasis/recurrence were retrospective/observational, study participant attrition was the main source of potential bias in this meta-analysis, either due to authors not reporting the number of cases lost to follow up or due to considerably uneven incidence of censoring in the cohorts. Another source of bias was the study confounding since authors did not report the tumor stage and comorbidities. In terms of inflammatory cytokines, the main limitation of the analysis was the small number of sample size in each cytokine. Meanwhile, no long-term outcomes (metastasis or recurrence) were recorded in those studies that failed to provide direct evidence between the perioperative inflammation and long-term outcomes for cancer. In addition, the severity of the surgical trauma varied between different studies, which may have significantly influenced the cytokine level and affected the efficacy of the general anesthesia.

## 5. Conclusions

Based on all the clinical and preclinical evidence, it is advisable to take precautions when administrating general anesthesia, especially inhaled anesthesia, to cancer patients. Important fundamental questions remain to be answered for the molecular targets of general anesthetics in cancer progress and metastasis. It is our opinion that the mechanistically defined effect of individual anesthetic could serve as convincing evidence on the anesthetic selection and help evaluate additional mitigating strategies during oncological surgery at early clinical stages or, hopefully, in the premetastatic phase. We urge more well-designed preclinical studies and perspective clinical trials with considerations of molecular subtypes to differentiate the adverse effects associated with the use of certain anesthetics.

## Figures and Tables

**Figure 1 cancers-15-02759-f001:**
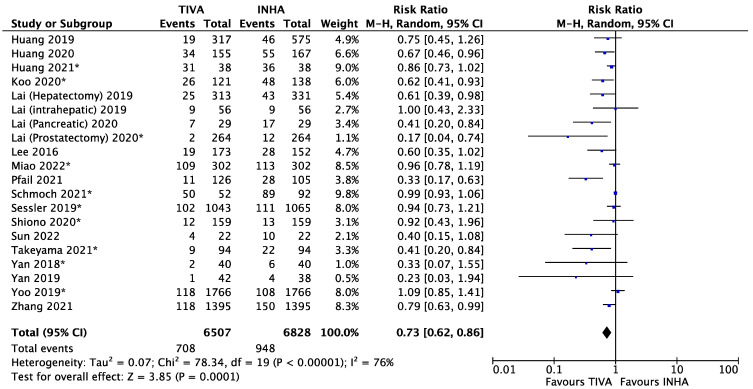
Forest plot of metastasis/recurrence of matched patients in TIVA and IHNA cohorts. The asterisk (*) refer to studies reporting recurrence.

**Figure 2 cancers-15-02759-f002:**
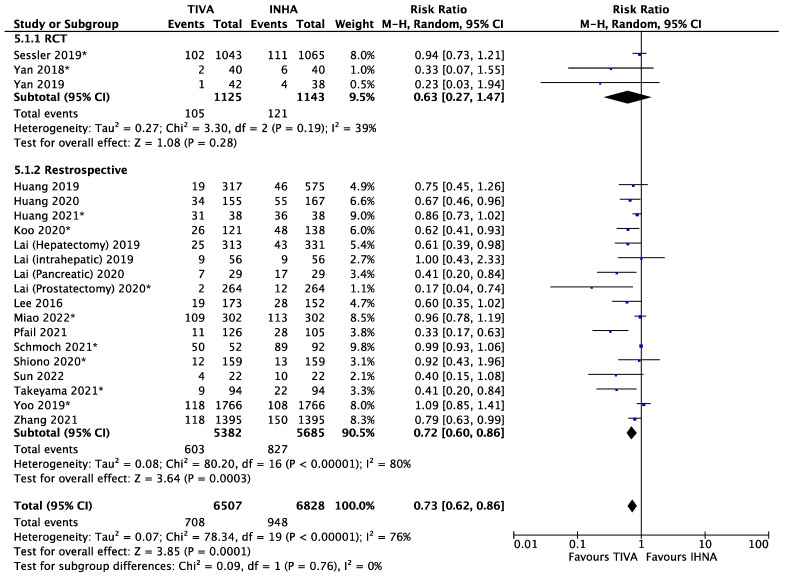
Subgroup analysis metastasis/recurrence of matched patients according to study type. The asterisk (*) refer to studies reporting recurrence.

**Figure 3 cancers-15-02759-f003:**
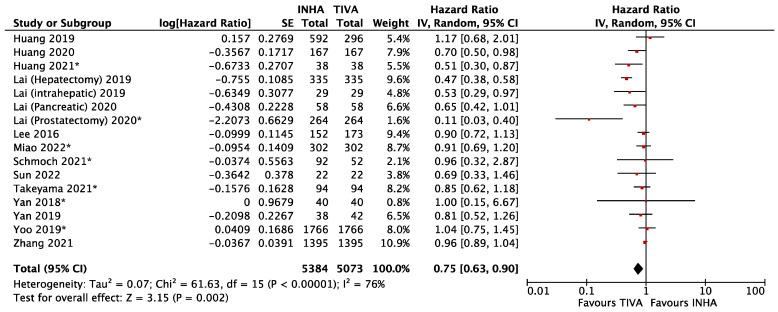
Forest plot of overall survival of matched patients in TIVA and IHNA cohorts. The asterisk (*) refer to studies reporting recurrence.

**Figure 4 cancers-15-02759-f004:**
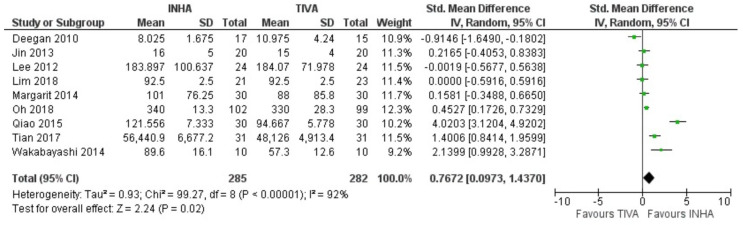
Forest plot of IL-6 at 24 h after cancer surgery in TIVA and IHNA cohorts.

**Table 1 cancers-15-02759-t001:** Meta-analysis search strategy and term.

Outcome	Anesthesia Term	Cancer Term
Metastasis/recurrence	TIVA/total intravenous anesthesia/propofol/inhaled anesthesia/volatile anesthesia	Cancer/malignancy/tumor/neoplasm
Cytokine/interleukin	TIVA/total intravenous anesthesia/propofol/inhaled anesthesia/volatile anesthesia	Cancer/malignancy/tumor/neoplasm

**Table 2 cancers-15-02759-t002:** Pre-clinical animal models studying the effect of general anesthetics on cancer metastasis.

STUDY	EXPERIMENTAL MODEL	CANCER TYPE	ANESTHESIA	SURGERY	OUTCOMES	RESULTS	MECHANISM
**FREEMAN 2019**	4T1 orthotopic breast cancer spontaneous metastasis mouse model	Murine 4T1 breast cancer	Sevoflurane vs. sevoflurane + propofol vs. sevoflurane + lidocaine	Y	Post-operative pulmonary and hepatic metastasis; serum VEGF and IL-6 level at final	Propofol and lidocaine reduced pulmonary metastasis; No difference for hepatic metastasis or serum IL-6, VEGF at the end of observation	
**LI 2020**	orthotopic breast cancer spontaneous metastasis mouse model	Human MDA-MB-231 breast cancer and murine 4T1 breast cancer	Sevoflurane vs. Propofol	Y	Post-operative pulmonary metastasis; post-operative IL-6	Surgery under sevoflurane significantly increased lung metastasis than with propofol; Sevoflurane increased serum IL-6 and infiltration of CD11b+ myeloid cells into lung	Sevoflurane induced pro-metastatic effects by activation of IL-6/STAT3 pathway and infiltrated CD11b+ cells.
**LIU 2021**	Experimental metastasis model	Human colorectal carcinoma	vehicle (DMSO) vs. propofol	N	Pulmonary metastasis formation	Propofol promote tumor metastasis to the lungs as compared to control	Propofol enhanced adhesion and extension of tumor cells to endothelial cells by activation of GABAAR-dependent TRIM21 modulation of Src expression
**LIU 2022**	4T1 orthotopic breast cancer spontaneous metastasis mouse model	Murine 4T1 breast cancer	Isoflurane vs. sevoflurane vs. desflurane	Y	Post-operative pulmonary metastasis; serum level of IL-6, CCL-1, MCP-1, and VEGF at final	No difference in pulmonary metastasis or inflammatory cytokines under different inhalational anesthetics	
**LU 2020**	Orthotopic tumor model and experimental metastasis model	Human T24 bladder cancer	control vs. isoflurane	N	Primary tumor growth; hepatic metastasis	Isoflurane exposure accelerated formation of primary tumor and hepatic metastases	Isoflurane promotes epithelial-mesenchymal transition and metastasis by HIF-1alpha-beta-catanin/Notch1 pathway
**MAMMOTO 2002**	Subcutaneous inoculation	Murine osteosarcoma	vehicle (DMSO) vs. propofol	N	Primary tumor growth; pulmonary metastatic nodule	No difference in primary tumor volume; Continuous infusion of propofol inhibited pulmonary metastasis of LM 8 cells in mice	
**MOUDGIL 1997**	Experimental metastasis model	Murine B16 melanoma	control vs. halothane vs. isoflurane	N	Pulmonary metastasis	More metastases were observed in animals’ exposure to halothane or isoflurane than in the control	

For surgery, Y for yes and N for n.

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
