# Peer review of "The Potential Effect of General Anesthetics in Cancer Surgery: Meta-Analysis of Postoperative Metastasis and Inflammatory Cytokines"

_cancers, 2023, doi:10.3390/cancers15102759_

Round 1

Reviewer 1 Report

Findings of the article

1.     Intravenous propofol anesthesia RR=.67 is a better option for onco-surgery over inhalation anesthesia RR= .73 as it has lower risk of recurrence.

2.     Breast cancer and liver patients (RR= .65) showed better performance post surgery in TIVA cohort as compared to INHA(RR=.87)

3.     Forest plot of 20 studies indicates that TIVA is favoured over IHNA for onco-surgery to reduce the risk of metastasis and recurrence.

4.     Forest Plot for overall survival also favours TIVA over IHNA (hazard ratio (HR) = 203 0.75, 95% CI 0.63 to 0.90, I2 = 76%, p = 0.002)

5.     The underlying cause of metastasis in case of IHNA was via IL-6 production which was significantly low in TIVA cohort with standardised mean difference (SMD) = 0.77, 221 95% CI = 0.10 to 1.44, I2 = 92%, p = 0.02) but there was no significant difference among each cancer types shown through subgroup analysis of cancer types.

Conclusion

This metanalysis tried to convey that TIVA over IHNA is better option for oncological-surgery as it has lower risk of recurrence and metastasis via decreased production of IL-6 and TNF-α known to be a pro-inflammatory cytokines.

Although, a more detailed and large cohort of patients is required to analyse all the subgroups included in the paper to create a clear path of understanding for choosing the type of anaesthesia as perioperative measure to lower the risk of recurrence and metastasis.

Shortcomings

1.     In method, the search strategy should be tabulated for better understanding.

2.   Method did not include formulas and derivations illustrations rather only names of theorems are stated.

3.Result should also include insights of molecular targets of these anesthesia to better validate the findings as to which pathways are triggered in case of IHNA leading to more risk of recurrence and metastasis which is opposite in case of TIVA.

Author Response

Reviewer 1

  1. In method, the search strategy should be tabulated for better understanding.

Thank you for this suggestion. We have included Table 1 to present search strategy as in p2-p3, line 88-99

  1. Method did not include formulas and derivations illustrations rather only names of theorems are stated.

The Mantel-Haenszel method to calculate relative risk (RR) and inverse variance method for calculate hazard ratio (HR) are built in Review Manager (RevMan). Both are standard methods to use and we did not make any derivations from the original formula.

  1. Result should also include insights of molecular targets of these anesthesia to better validate the findings as to which pathways are triggered in case of IHNA leading to more risk of recurrence and metastasis which is opposite in case of TIVA.

We would like to thank the reviewer for the valuable suggestion. Most of the clinical studies did not analyze the molecular mechanism. For that reason, we further assess the post-operative inflammatory cytokines as the potential targets of general anesthetics. We found that IHNA increased circulating IL-6 (Figure 3), which has been demonstrated to be correlated to cancer metastasis (Discussion, p9, line 396-403). The activation of IL-6 signaling pathway by IHNA was also observed in pre-clinical animal studies (Discussion, p9, line 411). However, the effect of general anesthetics on cancer metastasis is complex and may vary depending on the specific agent used, the duration and depth of anesthesia, the type and stage  of cancer, and other factors. Further research is needed to better understand the mechanisms underlying these effects and to develop strategies to minimize any potential harm.

Reviewer 2 Report

This is an interesting clinical question - namely, local intra-operative anesthetic choices which may impact cancer recurrence - which remains unresolved.  The authors have a done a nice job presenting a large clinical and pre-clinical meta-analysis.  However, in any such study there are inherent limitations that if commented upon may significantly improve the conclusions:

- There should be some comment as to why IHNA vs TIVA is typically chosen.  Is this typically local hospital/anesthesiologist preference?  Or are there specific patient factors that contribute?  If patients who receive IHNA are typically older, more frail, more advance disease, etc this may be masking underlying bias for why these patients had worse cancer outcomes.  The RTC in breast cancer perhaps illustrates this.  

- Are there any data regarding the timeline of recurrences of cancer? For example, if IHNA is associated not only with more recurrences but more early recurrences - each of the referenced cancer types should have an expected period of highest risk of recurrence - this could even further strengthen the clinical relevance of the paper.  

Author Response

Reviewer 2

- There should be some comment as to why IHNA vs TIVA is typically chosen.  Is this typically local hospital/anesthesiologist preference?  Or are there specific patient factors that contribute?  If patients who receive IHNA are typically older, more frail, more advance disease, etc this may be masking underlying bias for why these patients had worse cancer outcomes.  The RTC in breast cancer perhaps illustrates this.  

Thank you so much for the valuable suggestion. Both IHNA and TIVA are in the current standard care. The choice of general anesthetics is usually the decision of the anesthesiologist based on the co-morbidity and condition of patient, and the preference of the anesthesiologist. We have included this information in introduction p2 line51-52

- Are there any data regarding the timeline of recurrences of cancer? For example, if IHNA is associated not only with more recurrences but more early recurrences - each of the referenced cancer types should have an expected period of highest risk of recurrence - this could even further strengthen the clinical relevance of the paper.  

Thank you so much for the valuable insight. There recurrence and metastasis of cancer vary up to 20 years after surgery. For breast cancer for example, a substantial fraction of patients develop metastasis relatively soon after resection of their primary tumors, with a sharp rise begins 6 months after surgery and peaks 6 to 12 months later. The early relapses are conserved across subtypes of breast cancer. As suggested, we included the follow-up time for all the included clinical studies in supplementary Table 1.

Reviewer 3 Report

The authors presented a systematic review and meta-analysis of clinical and animal studies comparing overall survival, metastasis/recurrence rate, and selected cytokines levels after surgery between inhalational anesthesia and propofol-based total intravenous anesthesia. The study methodology is perfect, the analytical results are clearly reported, and the interpretation of the results is appropriately done with a conservative tone, given the heterogeneity of the study designs, populations, interventions and diagnoses, The only aspect I think needs clarification is that the authors did not present an evaluation of publication bias, although in the methodology section it was stated that publication bias would be evaluated with funnel plots and Egger's test.

Author Response

Thank you so much for the valuable suggestion. The egger’s test was performed for each analysis and presented by P value in each figure under “Heterogeneity”. We also included the funnel plots for studies involved in metastasis/recurrence analysis (Supplementary Figure 3). We also discussed all potential limitation at the end of discussion. All the changes are highlighted in grey.

Reviewer 4 Report

MS Number: cancers-2321735

MS Title: The potential role of general anesthetics in cancer surgery…

MS Authors: Li et al.

Date of Review: 20230418

This is an interesting paper. The authors conducted a systematic review and meta-analysis to examine the possible effects of general anesthetics on the cancer metastasis and recurrence after surgery. Also animal studies were included for analysis in this report.

Major issues:

l   The contents in the Results and Discussion sections are not written in a usual manner. For example, too much discussion in the Results section and too little in the Discussion to say about the inconsistent findings in the cited literature.

l   It will be helpful for the readers if the authors could discuss the opposite findings among each research groups (especially several citations were from the same research groups).

l   It will be helpful if the bias and inhomogeneity of the analysis could be discussed in more details.

Minor issues:

l   Title: The potential “role”: Will “effect, influence or impact” be more suitable?

l   Simple Summary: “Pre-clinical studies confirmed clinical observation and explored potential mechanism in a limited extent.” It sounds very difficult to confirm something if the argument is “in a limited extent”.  

l   Abstract:

n   “In conclusion, the current evidence suggests intravenous anesthetic propofol is associated with better outcome over inhaled anesthetics …” The “better outcome” was not defined here, e.g., long-term or short-term outcomes.

n   “Preclinical animal model studies show that inhalational anesthetics increase the risk of cancer metastasis compared to propofol.” Is this statement overstated because it apparently is generalizable for various kinds of cancers.

l   Keywords: More relevant keywords could be added.

l   Introduction:

n   Page 1, line 37: “Surgery is often the first-line treatment for millions of patients with solid tumors..” Please cite references for this statement. Treatment of solid cancers typically involves the use of multiple modalities, e.g., surgery, systemic anti-cancer therapy (SACT), and radiotherapy, alone or in combination or sequentially.

n   Page 2, line 54: “have significant effects on the biology of cancer cells and their role in clinical outcome” [ref 2-5] What kind of “effects” of anesthetics on cancer biology and possible clinical outcomes?

n   Page 2, lines 65 and 67: The citation of Ref 7 and Ref 8 should specifically be applied to breast and lung cancer, respectively (unless the findings of association are generalizable to other cancer types).

l   Discussion

n   Page 9, line 328: “Our meta-analysis showed that post-operative level of IL-6 and TNF-α were significantly lower in the propofol group, which indicate the anti-inflammatory effects of propofol.” However, in the Results section (Page 6, line 233) regarding the effects on IL-6, the authors stated “The quality of evidence was very low due to significant data heterogeneity, as well as high risk of publication bias.” Please clarify this kind of inconsistency.

n   Page 10, line 343: The third objective of this meta-analysis is about the overview of the pre-clinical studies (Table 2). The purpose of this overview was to get some potential molecular targets/mechanisms. However, as the Table 2 shows, ref Freeman 2019 and Liu 2020 provides no mechanisms (compared to that by Li 2020). Neither by ref Mammoto,2020 and Moudgil 1997. Do the Table 2 and the analysis (Page 7, line 255) and discussion (Page 10, line 343) provide any other supportive argument than the one from Ref 49 (e.g., by IL-6/JAK/STAT3 pathway or CD11b)? Namely, could the pre-clinical results support the findings (association) described in the Results 3-1?

n   Page 10, 364: In the Conclusion section, could this meta-analysis supprot the statement: “Based on all the clinical and preclinical evidence, we suggest that propofol based anesthesia is safer for oncological surgery.” ?

l   It is better for the authors to discuss the discrepancy of the study results from the Ref 32 (Yoo, 2019) and Ref 34 (Sessler, 2019)?

Author Response

Major issues:

  1. The contents in the Results and Discussion sections are not written in a usual manner. For example, too much discussion in the Results section and too little in the Discussion to say about the inconsistent findings in the cited literature. 

We would like to thank the reviewer for the valuable suggestion. We have rearranged the Results and Discussion sections as highlighted in green.

  1. It will be helpful for the readers if the authors could discuss the opposite findings among each research groups (especially several citations were from the same research groups).

Thank you so much for the suggestion. We have included those in discussion. The discussion of RCT from Sessler 2019 is on page 9 line 444-449 and discussion of preclinical study from Liu 2021 is on page 10 line 518-522.

  1. It will be helpful if the bias and inhomogeneity of the analysis could be discussed in more details.

Thank you so much for pointing out the missing information. We have included discussion on bias and heterogeneity at the end of discussion section (highlighted in grey)

Minor issues:

l   Title: The potential “role”: Will “effectinfluence or impact” be more suitable?

l   Simple Summary: “Pre-clinical studies confirmed clinical observation and explored potential mechanism in a limited extent.” It sounds very difficult to confirm something if the argument is “in a limited extent”.   

l   Abstract:

n   “In conclusion, the current evidence suggests intravenous anesthetic propofol is associated with better outcome over inhaled anesthetics …” ㄍThe “better outcome” was not defined here, e.g., long-term or short-term outcomes.

n   “Preclinical animal model studies show that inhalational anesthetics increase the risk of cancer metastasis compared to propofol.” Is this statement overstated because it apparently is generalizable for various kinds of cancers.

l   Keywords: More relevant keywords could be added.

l   Introduction: 

n   Page 1, line 37: “Surgery is often the first-line treatment for millions of patients with solid tumors..” Please cite references for this statement. Treatment of solid cancers typically involves the use of multiple modalities, e.g., surgery, systemic anti-cancer therapy (SACT), and radiotherapy, alone or in combination or sequentially.

n   Page 2, line 54: “have significant effects on the biology of cancer cells and their role in clinical outcome” [ref 2-5] What kind of “effects” of anesthetics on cancer biology and possible clinical outcomes?

n   Page 2, lines 65 and 67: The citation of Ref 7 and Ref 8 should specifically be applied to breast and lung cancer, respectively (unless the findings of association are generalizable to other cancer types).

l   Discussion

n   Page 9, line 328: “Our meta-analysis showed that post-operative level of IL-6 and TNF-α were significantly lower in the propofol group, which indicate the anti-inflammatory effects of propofol.” However, in the Results section (Page 6, line 233) regarding the effects on IL-6, the authors stated “The quality of evidence was very low due to significant data heterogeneity, as well as high risk of publication bias.” Please clarify this kind of inconsistency.

n   Page 10, line 343: The third objective of this meta-analysis is about the overview of the pre-clinical studies (Table 2). The purpose of this overview was to get some potential molecular targets/mechanisms. However, as the Table 2 shows, ref Freeman 2019 and Liu 2020 provides no mechanisms (compared to that by Li 2020). Neither by ref Mammoto,2020 and Moudgil 1997. Do the Table 2 and the analysis (Page 7, line 255) and discussion (Page 10, line 343) provide any other supportive argument than the one from Ref 49 (e.g., by IL-6/JAK/STAT3 pathway or CD11b)? Namely, could the pre-clinical results support the findings (association) described in the Results 3-1?

n   Page 10, 364: In the Conclusion section, could this meta-analysis supprot the statement: “Based on all the clinical and preclinical evidence, we suggest that propofol based anesthesia is safer for oncological surgery.” ?

l   It is better for the authors to discuss the discrepancy of the study results from the Ref 32 (Yoo, 2019) and Ref 34 (Sessler, 2019)?

Thank you so much for the detailed suggestion. All the corrections are made accordingly and highlighted in green.

Round 2

Reviewer 2 Report

The authors have adequately addressed my initial review, and I would recommend publication.  

Reviewer 4 Report

Dear Authors

Thanks for your detailed and careful response/revisions to my review comments. Now I felt comfortable to make recommendation to accept your paper to be published.